



# A transient CGCM simulation of the past 3 million years

Kyung-Sook Yun [1,2*], Axel Timmermann[1,2*], Sun-Seon Lee[1,2], Matteo Willeit[3], Andrey Ganopolski[3], and Jyoti Jadhav[1,2]

[1]Center for Climate Physics, Institute for Basic Science (IBS), Busan, Republic of Korea

[2]Pusan National University, Busan, Republic of Korea

[3]Potsdam Institute for Climate Impact Research, Potsdam, Germany

*Correspondence to*: Kyung-Sook Yun (kssh@pusan.ac.kr), Axel Timmermann (axel@ibsclimate.org)

**Abstract. Driven primarily by variations in earth's axis wobble, tilt, and orbit eccentricity, our planet experienced massive glacial/interglacial reorganizations of climate and atmospheric $CO_2$ concentrations during the Pleistocene (2.58 Ma – 11.7 ka). Even after decades of research, the underlying climate response mechanisms to these astronomical forcings have not been fully understood. To further quantify the sensitivity of the earth system to orbital-scale forcings we conducted an unprecedented quasi-continuous coupled general climate model simulation with the Community Earth System Model version 1.2 (CESM1.2, ~3.75° horizontal resolution), which covers the climatic history of the past 3 million years ago (3 Ma). In addition to the astronomical insolation changes, CESM1.2 is forced by estimates of $CO_2$ and ice-sheet topography which were obtained from a simulation previously conducted with the CLIMBER-2 earth system model of intermediate complexity. Our 3 Ma simulation consists of 42 transient interglacial/glacial simulation chunks, which were partly run in parallel to save computing time. The chunks were subsequently merged, accounting for spin-up and overlap effects to yield a quasi-continuous trajectory. The computer model data were compared against a plethora of paleo-proxy data and large-scale climate reconstructions. For the period from the Mid-Pleistocene Transition (MPT, ~1 Ma) to the late Pleistocene we find good agreement between simulated and reconstructed temperatures in terms of phase and amplitude (−5.7°C temperature difference between Last Glacial Maximum and Holocene). For the earlier part (3 Ma – 1 Ma), differences in orbital-scale variability occur between**





model simulation and the reconstructions, indicating potential biases in the applied $CO_2$ forcing. Our model-proxy data comparison also extends to the westerlies, which show unexpectedly large variance on precessional timescales, and hydroclimate variables in major monsoon regions. Eccentricity-modulated precessional variability is also responsible for the simulated changes in the amplitude and flavours of the El Niño-Southern Oscillation. We further identify two major modes of planetary energy transport, which played a crucial role in Pleistocene climate variability: the first obliquity and $CO_2$-driven mode is linked to changes in the equator-to-pole temperature gradient; the second mode regulates the inter-hemispheric heat imbalance in unison with the eccentricity-modulated precession cycle. During the MPT, a pronounced qualitative shift occurs in the second mode of planetary energy transport: the post-MPT eccentricity-paced variability synchronizes with the $CO_2$ forced signal. This synchronized feature is coherent with changes in global atmospheric and ocean circulations, which might contribute to an intensification of glacial cycle feedbacks and amplitudes. Comparison of this paleo-simulation with greenhouse warming simulations reveals that for a RCP8.5 greenhouse gas emission scenario, the projected global mean surface temperature changes over the next 7 decades would be comparable to the late Pleistocene glacial-interglacial range; but the anthropogenic warming rate will exceed any previous ones by a factor of ~100.

## 1 Introduction

Glacial cycles during the Pleistocene (2.58 million years ago (Ma) – 11.7 thousand years ago (ka)) were characterized by global mean surface temperature (GMST) swings that attained values of up to ~ 6 °C (Schneider von Deimling et al., 2006;McClymont et al., 2013;Tierney et al., 2020). These variations in temperature and the associated waxing and waning of continental ice-sheets were accompanied by relatively small changes ($< \sim 0.5$ W m$^{-2}$) in earth's global mean radiation balance (Baggenstos et al., 2019), but considerable variations in the seasonal and latitudinal distribution of insolation (Masson-Delmotte et al., 2013). Transient coupled atmosphere/ocean simulations either with earth system models



of intermediate complexity (EMICs) (Menviel et al., 2011;Timm and Timmermann, 2007;Timm et al., 2008;Timmermann et al., 2009;Timmermann et al., 2014a;Timmermann and Friedrich, 2016) or based on coupled general circulation models (CGCMs) (Timmermann et al., 2022;Liu et al., 2009;Singarayer and Valdes, 2010;Zhang et al., 2021;Timmermann et al., 2007) have helped elucidate how orbital-scale forcings translate into global and regional climate responses. Transient orbital-scale ice-sheet sensitivities and feedbacks with the atmosphere and ocean have so far only been explored with intermediate complexity models (Ganopolski et al., 2010;Willeit and Ganopolski, 2018;Willeit et al., 2019;Heinemann et al., 2014) or forced offline ice-sheet models with various representations of climatic feedbacks (Tigchelaar et al., 2018;Tigchelaar et al., 2019;Abe-Ouchi et al., 2013).

To gain a better understanding of how glacial/interglacial variability emerged during the Pleistocene, it has been beneficial to study the lateral flow of energy in our climate system. Orbital-scale shifts in atmospheric and oceanic heat transports influence key atmospheric circulation features such as the Hadley circulation, the Inter-tropical Convergence Zones (ITCZs), and midlatitude storm tracks (Liu et al., 2017;Liu et al., 2015;Chiang and Bitz, 2005;Kaspar et al., 2007). Previous modeling studies (Timmermann et al., 2014a;Liu et al., 2017;Kamae et al., 2016;Donohoe et al., 2020;Kim, 2004) have elucidated the relationship between external forcings, climate feedbacks, and lateral energy redistributions by focusing on specific Pleistocene periods, such as the Last Glacial Maximum (LGM) or extreme orbital conditions. However, little is known on how the interplay between orbital and greenhouse gas forcing (GHG) has shaped the meridional transport of heat in the atmosphere and ocean over the past 3 million years and whether these two components reinforced or compensated each other (Bjerknes, 1964;Stone, 1978). To understand how Milanković forcing generates glacial variability, it is necessary to determine how changes in atmospheric and oceanic heat transports feed back to the climate mean state. This is particularly interesting in the context of the Mid-Pleistocene Transition (MPT; ~ 1.25-0.7 Ma) when the dominant periodicity of ice-volume and global mean temperature changed from 41 kyr to 80-120 kyr cycles (Clark et al., 2006;Bintanja and van de Wal, 2008;McClymont et al., 2013).

To address these fundamental questions, we conducted an unprecedented transient CGCM simulation that covers the climate history of the past 3 Ma. This quasi-continuous model simulation,



represents a 1-million-year extension of the 2 Ma Community Earth System Model (CESM) simulation,
which has originally been conducted to study how climate shaped early human habitats (Timmermann et
al., 2022). The 2 Ma and 3 Ma simulations are based on the CESM version 1.2 in 3.75° atmospheric and
nominal 3° oceanic horizontal resolutions (T31_gx3v5). Our CESM1.2 3 Ma simulation represents the
first full Pleistocene transient model simulation conducted with a 3 dimensional CGCM to date.

This paper describes the overall performance of the 3 Ma climate simulation and highlights
important processes related to changes in Pleistocene global heat transport, feedbacks and variability. The
paper is organized as follows. Section 2 introduces the model set-up and experimental design. In Section
3, we compare the simulation with proxy data for mean temperature, hydroclimate, and large-scale
westerly winds. Section 4 illustrates how the large-scale external forcings can influence modes of natural
climate variability, exemplified here by the El Niño-Southern Oscillation (ENSO) phenomenon. We then
investigate the global energy transport changes in Section 5 and the related large-scale atmosphere and
ocean changes in Section 6. Section 7 discusses the main results and provides a future outlook.

## 2 Model set-up and experimental design

Our transient Pleistocene paleoclimate model simulation uses the coupled CESM version 1.2
(Hurrell et al., 2013), with CAM4 atmospheric physics (T31, ~3.75° resolution, 26 levels), the POP2
ocean model (gx3v5, 100×116 horizonal grids and 25 vertical levels), CLM4.0 land physics (prescribed
vegetation), and CICE4 (Los Alamos Sea Ice Model) sea ice components. Unlike previous transient EMIC
simulations (Willeit et al., 2019;Timmermann et al., 2014a;Ganopolski et al., 2010;Goosse et al., 2010),
the CESM1.2 simulation includes more realism and complexity, e.g., interactive clouds, a realistic
representation of tropical processes, a 3-dimensional ocean circulation, and a fully dynamic atmosphere.
We also split the simulation into 42 chunks (minimum 32 kyr and maximum 125 kyr length) which
overlapped by about 5,000 years (see Fig. 1a), and ran them in parallel on the ICCP/IBS supercomputer
*Aleph*. Each chunk was initialized with peak interglacial conditions (maximum boreal summer insolation;
e.g., 125 ka, 243 ka, and so on) using the same initial and restart data from a preindustrial 2,300-year-



long spin-up control simulation. We finally obtained the "3 Ma" full history of climate by concatenating the data from each chunk and by applying a time-sliding linear interpolation for the overlap periods (Timmermann et al., 2014a). By using a relatively small acceleration (factor 5 for orbital, GHG, and ice sheets forcings) (Timmermann et al., 2014a;Lorenz and Lohmann, 2004), we ensure that even the deep ocean with advective adjustment and tracer timescales of a few thousand years will be close to quasi-equilibrium with the fastest "accelerated" forcing timescale (Timm and Timmermann, 2007) – i.e. the precessional cycle. This acceleration method of factor 5 reduces the 3,000,000 orbital year forcing to 600,000 model years in CESM.

The CESM uses the constant LGM bathymetry and land-sea mask (i.e., 21 ka) based on ICE-6G_C data (Argus et al., 2014). Figure 1a displays the temporal evolution of three transient climate forcings of summer insolation at 65°N, $CO_2$, and global ice volume in sea level equivalent (Willeit et al., 2019). Overall, the interglacial peaks (initial time at each ensemble simulation, colored dots in Figs. 1b-c) are characterized by high $CO_2$ concentrations and low ice volume, with the orbital phase being less constrained (Figs. 1b-c). In our 3 Ma experiment, we updated the forcings of GHG concentrations (Lüthi et al., 2008;Willeit et al., 2019;Lisiecki and Raymo, 2005), Northern Hemisphere (NH) ice-sheet extent and topography (Willeit et al., 2019), and astronomical insolation variations (Berger, 1978) every 100 orbital years (i.e., 20 model years). For GHGs, we combined different datasets: after 800 ka, measured air-bubble $CO_2$ concentrations from the European Project for Ice Coring in Antarctica (EPICA) Dome C ice core (Jouzel et al., 2007) were used; prior to 800 ka, we used the simulated $CO_2$ from the CLIMBER-2 transient simulation (Willeit et al., 2019), which includes an interactive carbon cycle. We also included estimates of the $CH_4$ and $N_2O$ concentrations as a CESM forcing using a regression of the EPICA $CH_4$ and $N_2O$ concentrations with the global benthic $\delta^{18}O$ stack (Lisiecki and Raymo, 2005) (post 800 ka) and extended this linear statistical model to 3 Ma using the benthic $\delta^{18}O$ stack data (see Fig. S1 in the Supplement). The ice mask and ice-sheet topography changes were also obtained from the simulated CLIMBER-2 orography data (Willeit et al., 2019). For orbital forcing, it is worth mentioning that before the past 1 Ma, the solutions of Berger (1978) are quite different from those of Laskar et al. (2004). This discrepancy in pre-MPT orbital forcing and the consequent changes in climate system would be further





checked in future work. CESM1.2 in our resolution has a relatively weak climate sensitivity of 2.4 °C warming for a doubling relative to pre-industrial $CO_2$ levels. This is about 1.5–fold smaller than the
Coupled Model Intercomparison Project Phase 6 (CMIP6) multi-model ensemble-based estimation (3.7 ± 1.1 °C within 37 multi-models) (Meehl et al., 2020). To compensate for this relatively weak climate sensitivity of CESM1.2 and to indirectly account for the effect of unresolved and highly uncertain dust forcing (Friedrich et al., 2016;Kohler et al., 2010), which is also correlated with the benthic $\delta^{18}O$ stack, we re-scaled the GHGs forcing anomalies relative pre-industrial conditions by a factor of 1.5.

## 3 Model-proxy comparison

### 3.1 Climate sensitivity and polar amplification

An important question to address is whether the 3 Ma simulation reproduces the magnitude and
timing of glacial/interglacial variability during the Pleistocene. We first compare results of our 3 Ma simulations with a variety of long-term paleo-proxy data and the CLIMBER-2 simulation. Figure 2a illustrates the model's capability in capturing the magnitude and timing of GMST variations, shown by previous model- and proxy-based estimates (Willeit et al., 2019;Friedrich et al., 2016). Our simulation exhibits a −5.7 °C glacial cooling at the LGM (~21 ka) relative to pre-industrial conditions (~ 0 ka). This
magnitude of cooling agrees well with a combined proxy-model estimate of -6.3 to -5.6 (95% confidence interval) (e.g., Tierney et al., 2020). Simulated sea surface temperatures (SSTs) also reveal a high degree of coherence for the post MPT period with proxy-based SST reconstructions in the tropics (Herbert et al., 2010) (Fig. 2b) and elsewhere (Petrick et al., 2019;Cartagena-Sierra et al., 2021;Lawrence et al., 2009;de Garidel-Thoron et al., 2005;Russon et al., 2010) (Figs. 2c-g), indicating a reliable simulation of spatial
SST patterns in addition to the temporal evolution. However, prior to the MPT, the model simulation diverges considerably from some of the SST proxies in terms of representing the exact orbital phase, as well as the long-term extratropical gradual Pleistocene trends. For instance the tropical SST stack (Herbert





et al., 2010) exhibits higher pre-MPT variability on obliquity timescales of 41 kyrs (Fig. 3a), as compared to the 3 Ma simulation, which features a more pronounced precessional cycle (~21 kyrs) (Fig. 3b). This mismatch might be in part due to an unrealistic orbital-scale variability (e.g., presence of precession) of the pre-MPT $CO_2$ forcing (Fig. 3c) or potential seasonal biases of some of the SST proxies (Timmermann et al., 2014b).

Various temperature reconstructions and simulations reveal amplified Arctic/Antarctic surface air temperature changes on glacial/interglacial timescales relative to the global average (Stap et al., 2018). This polar amplification is very pronounced in temperature reconstructions based on the EPICA Dome C ice core (Jouzel et al., 2007) and North Greenland Ice Core Project (NGRIP) (Barker et al., 2011), with Greenland's amplification being about twice as strong as Antarctica's, as documented by the LGM cooling of −18.2 °C for Greenland and −9.5°C for Antarctica (Fig. 4). Stronger polar amplification in Greenland can be partly explained by a larger lapse rate feedback (Hahn et al., 2020) and local impacts from the extensive NH ice-sheets (Smith and Gregory, 2012). The 3 Ma simulation also captures the difference in polar amplification between Greenland and Antarctica: the amplitude of LGM cooling attains values of ~ −15.7 °C for Greenland (75°N, 42°W) and −8.6°C for Antarctica (75°S, 123°E). The simulation also captures well the rapid trends of deglacial warming in both NH and Southern hemisphere (SH), which can be explained by a similar-shaped GHG. Overall, temperature reconstructions from Greenland and Antarctica (Jouzel et al., 2007;Barker et al., 2011;Kindler et al., 2014) correlate well on orbital timescales with the 3 Ma simulation. These results reflect the high fidelity of the simulations's climate sensitivity on global and regional scales and its response to the variety of acting forcings. Our simulation also provides a model-based glimpse into what temperature variability to expect in ongoing ice-core projects (Lilien et al., 2021) that plan to retrieve Antarctic ice, which is significantly older than the oldest EPICA ice.

## 3.2 Hydroclimate and large-scale westerly variability

Apart from the mean temperatures, orbital-scale changes in large-scale temperature gradients also play a crucial role in driving anomalies in hydroclimate (Schneider et al., 2014) and the extratropical atmospheric circulation (Timmermann et al., 2014a). Here we compare simulated changes in precipitation and atmospheric westerlies with proxy-data that show strong sensitivity to these factors.

As the rising branch of the Hadley circulation, the ITCZ is characterized by a belt of deep convective precipitation near the equator. Paleoclimate studies suggest that the latitudinal migration of the ITCZ was mainly driven by the precessional cycle (Schneider et al., 2014;Clement et al., 2004;Davis and Brewer, 2009;Liu et al., 2015). For low precession values (i.e., Northern summer perihelion), NH summers receive more insolation, which leads to summer warming and a northward migration of the ITCZ and Hadley circulation (e.g., Kang et al., 2018;Tigchelaar and Timmermann, 2016). Our simulation reaffirms the precession-driven ITCZ movement, which is further amplitude-modulated by the 100 and 405 kyr eccentricity cycles, in agreement with some proxy records from tropical South America and Africa (Fig. 5). Our simulation emphasizes the importance of the 405 kyr eccentricity cycle in tropical hydroclimate, which has received less attention in previous studies. Since precipitation is a positive definite non-Gaussian climate variable, the 100 kyr and 405 kyr amplitude modulations of the precessional cycle (Fig. 5) also rectify into a long-term mean signal (as illustrated here for Cariaco Basin precipitation; orange line in Fig. 5c) with the same frequencies, thereby introducing eccentricity as a mean forcing into the climate system.

The seasonal migration of the Hadley cell is also linked to shifts in the global monsoon precipitation (An et al., 2015;Schneider et al., 2014). With the precessional cycle modulating the amplitude of the seasonal solar forcing, monsoon systems, such as the Indian, East Asian, African, and Australia monsoons, thus show dominant variability on precessional timescales (Fig. 6). Previous studies demonstrated that the past global monsoon precipitation variabilities were controlled by the orbitally-driven land-sea thermal contrast, meridional pressure gradients, atmospheric $CO_2$ concentrations, and the growth of NH ice sheet (Clemens and Prell, 2003;Clark et al., 2006;Kutzbach and Guetter, 1986). Although the global monsoons share similar dynamics and behaviors, there are distinct features of regional monsoon systems that need to be considered. For example, the proxy-inferred western Australia monsoon precipitation was weakened, and the amplitude of variability was strengthened during the late





Pleistocene (Fig. 6a). The amplitude of observed East Asian summer monsoon variability increased from the early (~2.7 Ma to 1.2 Ma) to the late Pleistocene (after 1.2 Ma) (An et al., 2015) (Fig. 6c). Whereas the reconstructed western Australian and East Asian summer monsoon proxies show an intensification in

amplitude after the MPT, this trend in variability is absent in the reconstructed Indian summer monsoon (Fig. 6b) and African monsoon (Fig. 6d) proxies. The situation is different for the 3 Ma simulation, where we see an intensification in variability for all monsoon systems, except for Africa. This intensification in the model is likely brought about by the intensifying amplitudes of ice-sheet and $CO_2$ forcings and the corresponding impacts on NH hydroclimate.

We next compare the simulated large-scale westerly wind changes with dust and marine productivity reconstructions from ice cores and marine sediment proxies. Figure 7 displays the simulated westerly wind changes in Pacific and Atlantic Ocean of both NH and SH. We can see clear precessional and obliquity-scale variability in the simulation, consistent with some wind-sensitive climate and biogeochemical proxies (Lamy et al., 2019;Naafs et al., 2012). For example, the regional characteristics

such as strong obliquity response in North Atlantic (Fig. 7b) and a precession signal in Southern Pacific (Fig. 7d) are captured qualitatively in the model. In our model simulation, there is a pronounced strengthening in the variability of the westerly jet stream after the MPT, which is attributable to the larger glacial cycle amplitude (Fig. 2a) and polar amplification (Fig. 4) and resultant stronger differences in the meridional temperature gradient. Overall, the agreement between model-simulated variability and the

proxy record further supports the fidelity of our modeling approach. A more in-depth regional comparison between climate model and proxies, which would account for proxy uncertainties etc., is beyond the scope of our presentation paper.

## 4 Changes in ENSO

In contrast to previous long-term transient modeling studies conducted with EMICs (Timmermann and Friedrich, 2016;Willeit et al., 2019), the CESM1.2 is capable of simulating ENSO variability and associated air-sea coupling process well, even in the coarse resolution adopted here (Liu et al., 2014) –





albeit with weaker amplitude as compared to the observations. To take a broad view of ENSO's sensitivity to external forcings over the past 3 Myr, we first show the entire monthly timeseries of Niño 3 index which is defined as 1.5-7-year band-pass filtered (Liu et al., 2014) Niño 3 SST (5°S-5°N 150°W-90°W) (see Fig. S3 in Supplement). It includes 7.2 million monthly values. A Morlet wavelet spectrum (Torrence and Compo, 1998) of the Niño 3 SST timeseries (including annual cycle) (Fig. 8) reveals strong precessional-scale amplitude modulations for the annual time scale variance (i.e., averaged wavelet variance in the 0.5-1.5 yr band), in agreement with recent studies (Timmermann et al., 2007;Lu et al., 2019). The annual cycle of Niño 3 SST tends to vary ~90° out of phase with the precession index (see Fig. 5b), whereas the interannual (2-8 yr) time scale wavelet variance of Niño 3 SST is in phase with precession index (Karamperidou et al., 2020). This creates an interesting dynamic, in which variance shifts from interannual to annual timescales in response to the precessional cycle.

Furthermore, we characterize, eastern Pacific (EP) and central Pacific (CP) ENSO flavors as the standard deviation of the Niño 3 and Niño 4 (5°S-5°N, 160°E-150°W) SST indices in a 1000-year window, respectively (Figs. 9a-b). We find that on precessional timescales EP ENSO variability is in phase with CP ENSO variability and CP ENSO variability on average tends to be larger than EP ENSO variability (i.e., variability ratio shown in Fig. 9c is less than 1.0). The ratio of EP to CP ENSO variability is an indicator for how ENSO flavors change in response to orbital forcing. The ratio exhibits – apart from the precessional cycle - a strong eccentricity component with ~100 kyr and ~405 kyr periodicities. The timing of the largest ratio (ratio > 2.0 and EP ENSO variability > 0.6 °C) occurs in ~1470 ka, ~1350 ka, ~970 ka, ~230 ka, and ~130 ka. The timing in the occurrence of ENSO flavors in CESM1.2 relative to the phase of the precessional cycle is similar to that described in recent studies (e.g., Karamperidou et al., 2015). Contrary to ENSO's precessionally-dominated variability, the mean state of equatorial EP SST is largely controlled by atmospheric $CO_2$ variability (Fig. 9d). The different orbital characteristics of both mean state and variability suggest that ENSO's response to Milanković cycles can be attributed to the modulation of the seasonal cycle rather than to changing instabilities of air-sea interactions with respect to the background state.





## 5 Global heat transport

Large-scale insolation and meridional temperature gradients are the main drivers for the global transport of energy. The 3 Ma simulation provides a unique opportunity to study the dominant modes of atmospheric and oceanic energy transport and their responses to orbital-scale forcings. To this end we calculated the meridional heat transport (MHT) at each latitude ($\theta$), based on an energetic framework (Donohoe et al., 2020).

$$MHT(\theta) = -2\pi a^2 \int_{\theta}^{90} \cos(\theta)\,[ASR(\theta) - OLR(\theta)]d\theta, \qquad (1)$$

where ASR represents the zonal mean absorbed solar radiation and OLR denotes the zonal mean of outgoing long-wave radiation at top-of-atmosphere (TOA). The non-zero global mean value of ASR – OLR were removed from the calculation to attain energy conservation at both poles (Donohoe et al., 2020). The oceanic MHT (OHT) is directly deduced from the advection of the Eulerian mean circulation and meso-scale eddies (i.e., bolus circulation + diffusion) – the latter of which were parameterized but not explicitly resolved. There is also an indirect approach to calculate OHT using surface heat fluxes. A recent study Yang et al. (2015) documented the consistency between the heat transports obtained from direct and indirect approaches in the CESM. The atmospheric MHT (AHT) is then calculated as the residual of the MHT determined from TOA radiation and the OHT derived from CESM direct output. This residual approach was also applied successfully in previous studies (Donohoe et al., 2020).

Figure 10a displays the patterns of total MHT, AHT, and OHT averaged over the past 3 Ma (solid lines). In the tropics, the MHT has strong contributions both from the atmosphere and ocean. The main sources are the atmospheric Hadley circulation and the wind-driven oceanic shallow overturning circulation in the Pacific and the deep meridional overturning circulation in the Atlantic (Schneider, 2017;Held, 2001). The poleward transport outside the tropics is mostly caused by the atmospheric eddy activity (i.e., transient eddies and stationary eddies) (e.g., Donohoe et al., 2020). Despite the coarse horizontal resolution, the CESM1.2 captures the MHT in both atmosphere and ocean realistically (Yang et al., 2015). To further illustrate that our coarse-resolution CESM1.2 version can also capture the present-day partitioning between AHT and OHT, we compare the simulated MHTs with recent observational





estimates (see Fig. A1 in Donohoe et al. (2020)). The comparison reveals an overall agreement in terms of amplitude and peak latitude; however, the model exhibits a stronger AHT and weaker OHT than the observations, in agreement with previous studies (Donohoe et al., 2020;Yang et al., 2015).

In the following we further explore the linkage between changes in the MHT (Fig. 10) and external forcings (precession, obliquity, ice sheet, and $CO_2$). The time-latitude changes of anomalous MHT over the past 3 Ma (Fig. 10c) show a complex spatio-temporal pattern which documents that all forcings play an important role in controlling the MHT: high obliquity values weaken the equator-to-pole temperature gradient (Timmermann et al., 2014a) and thus affect the mid-latitude MHT, in particular prior to the MPT. Precession has a stronger impact in the tropics and NH mid-latitudes, due to the seasonally asymmetric

continental responses to the precessional forcing (Merlis et al., 2013;Tigchelaar and Timmermann, 2016). The NH ice sheets alter the surface albedo and topography, thus changing the NH mid latitude stationary and transient eddy activities (Donohoe and Battisti, 2009), whereas decreasing $CO_2$ concentrations enhance the southward transport of net surface heat flux toward the high latitudes of SH Ocean (Kim, 2004).

The combined effect of these forcings leads to complex variations of the MHT by ~ 10% (0.51 PW for maximum – minimum, 1 PW = $10^{15}$ W) relative to the latitudinal maximum value of ~5.3 PW (see Fig. 10a). We can also see that the partitioning of the total MHT into atmospheric and oceanic contributions is time-dependent (Donohoe et al., 2020;Yu and Pritchard, 2019;Newsom et al., 2021;Liu et al., 2017). We find an increasing contribution of the Southern Ocean to the total MHT variability after

the MPT (~ 1Ma) due to the enhanced $CO_2$ variability (Fig. 10b), which indicates that the MPT caused the shifts in the global energy distribution and corresponding changes in regional climate feedbacks.

To further simplify the complex MHT dynamics shown in Fig. 10c, we perform an empirical orthogonal function (EOF) analysis of the zonally averaged MHT anomalies over the past 3 Myr. The principal component (PC) time series (Figs. 11a-b) of the two dominant MHT modes explain ~ 92% of

the total variance. PC1 is characterized by a strong 41ky obliquity cycle (correlation coefficient $r$ ~ 0.7 with obliquity timeseries) as well as a negative trend throughout the Pleistocene, whereas PC2 is more related to higher frequency variability associated with the eccentricity-modulated precession cycle (~ 21 kyr cycle; see Fig. S4 in Supplement). The PC1 variability also shows a somewhat stronger 100 kyr





cyclicity after the MPT. PC2 changes its overall character after the MPT, when precessional variability

with an eccentricity-modulated envelope (early Pleistocene) transitions into variability which has strong

eccentricity and $CO_2$ periodicities in the mean value (middle to late Pleistocene). We can reconfirm these

characteristic changes by comparing the simulated PC2 time series with reconstructed variabilities using

the precession and $CO_2$ forcing only (see Fig. S5 in Supplement). The post-MPT shift in PC1 and PC2

can be attributed mostly to the increase in the amplitude of $CO_2$ radiative forcing and its cyclicity (see

Fig. 3c), both of which were generated from the orbitally-forced climate-carbon cycle dynamics simulated

by the EMIC CLIMBER-2 (Willeit et al., 2019).

The corresponding dominant MHT patterns are obtained by regressing the MHT anomalies

against PC1 and PC2 (Figs. 11c-11d). The PC1-related net radiation flux anomalies at TOA and surface

(Fig. 11c) show a weakening of the equator-to-pole insolation gradient (less heat in low-latitudes and

more heat in high-latitudes for positive PC1 values), consistent with high obliquity forcing and the effect

of high $CO_2$ on polar amplification. The PC2-related net radiation flux anomalies (Fig. 11d) show a

pronounced interhemispheric radiation gradient (i.e., warmer SH and colder NH for positive PC2). To

compensate for the meridional differences of the PC1-related weakening of the equator-to-pole net

radiation gradient (Fig. 11c), an overall weakening of the poleward heat transport (i.e., anomalous

northward in SH and southward in NH, Fig. 11e) is generated; the PC2-related strengthening of inter-

hemispheric TOA gradient (Fig. 11d) in turn drives an anomalous northward heat transport (Fig. 11f).

From these results, we define the two energy transport modes as (i) tropical heat "convergence (or

divergence) mode" (PC1) and (ii) (northward or southward) "inter-hemispheric shift mode" (PC2),

respectively. The Pleistocene global energy transport is made up by the different combinations of these

two energy transport modes. The underlying processes could be accompanied by characteristic changes

in both atmosphere and ocean circulations driven by NH ice-sheet albedo, sea ice albedo, and surface

wind changes. The key difference between these two MHT mechanisms is associated with the orbital

configuration and the response to the NH ice sheet.



# 6 Large-scale atmosphere and ocean circulation changes

## 6.1 Atmosphere

The Northern Annular Mode (NAM) and the Southern Annular Mode (SAM) are fundamental patterns of large-scale atmospheric circulation variability in both NH and SH. The positive phase is related to the anomalous low pressure over the polar region and high pressure over the mid-latitudes, accompanied by the strengthened (and sometimes poleward-shifted) westerly wind and storm tracks (e.g., Screen et al., 2018). We examine the variations in NAM and SAM as key indicators for large-scale extratropical circulation changes. Figure 12 shows a strong correlation between NAM and SAM on orbital and super-orbital timescales (correlation coefficient $r \sim 0.79$ during 3 Myr) (Fig. 12a). Both show a strong amplitude on precessional timescales before the MPT (Figs. 12b and c). However, an important difference between NAM and SAM appears after the MPT. The post-MPT NAM has a broader spectrum band from precession (~21 kyr), obliquity (~41 kyr) to eccentricity (~100 kyr), which is compared to the precession-only variability of SAM. This NAM-SAM difference may be due to the existence of NH landmass and post-MPT ice sheet growth influencing extratropical stationary waves (Yin et al., 2008).

The changes in NAM than SAM are better correlated to the major two modes of global energy transport shown in Fig. 11 (particularly for PC1-NAM, $r \sim 0.79$ during 3 Myr; $r \sim 0.44$ for PC1-SAM). For positive PC1 and negative PC2 values, the orbitally (high obliquity and low precession) driven increase of incoming insolation in high-latitudes would have decreased NH ice sheet volume (not interactively simulated here but included implicitly in the tendency of the ice-sheet forcing). The retreat of ice sheets can weaken the large-scale circulations and stationary eddies over the NH mid-latitudes by changes in surface albedo and ice-sheet topography (Yamanouchi and Charlock, 1997). The weakening of stationary eddies leads to the reduced poleward (anomalous southward) AHT, especially in the NH mid-latitudes (see Fig. 11e). This would imply a central role of the NH large-scale circulation in modulating the global energy transport during the Pleistocene.





## 6.2 Ocean

Previous studies highlighted the importance of the Atlantic Meridional Overturing Circulation (AMOC) in global energy transport (Frierson et al., 2013;Yu and Pritchard, 2019). To assess the role of ocean circulation changes in our transient simulation, we investigate the variability of the AMOC (Fig. 13a). The AMOC strength, calculated here as the maximum meridional streamfunction below 500m and north of 28°N, shows increased variability after the MPT on precessional and obliquity timescales (Fig. 13b). Our simulation only shows weak variability on the 80−120 kyr frequency, indicating a relatively minor contribution from $CO_2$ and ice-sheet forcing on AMOC variability. It should be noted, however, that since our model does not include explicit meltwater forcing or calving from the ice-sheets, the simulated AMOC variability on millennial and orbital timescales will be unrealistically small. In our simulation the AMOC amplitude varies between a minimum of 12.6 Sv to a maximum of 27.7 Sv. This result illustrates that in the absence of ice-sheet freshwater forcing, orbital forcing can play a more important role in driving an AMOC variability as compared to the GHG forcing alone (Fig. 13c). More specifically the AMOC strength is strongly controlled by the obliquity and precession cycles associated with the two modes of global energy transport (Fig.11). As described in Section 5, for a positive PC1 we find an orbitally driven weakening of the atmospheric circulation and heat transport (Fig. 11e) which also leads to a reduction in surface winds. In turn this process leads to a weakening in the NH ocean gyre circulation during high obliquity (positive PC1 in Fig. 11a). High precession values corresponding to a positive PC2 (Fig. 11b) are associated with an overall stronger AMOC and poleward heat transport (Fig. 11f). Therefore, the complex interplay between PC1 and PC2 of MHT is characterized by a modulation in the strength of the wind-driven and thermohaline ocean circulation, and corresponding changes in MHTs.

## 6.3 Sea ice





To further illustrate the impact of external forcings on climate conditions, we focus on sea-ice. We find a very high correlation between NH sea ice and SH sea ice (r ~ 0.9) (Fig. 14a). But the variance of NH sea ice extent (standard deviation of ~ 0.16 $10^7$ km$^2$) is considerably smaller than that of SH sea ice (standard deviation of ~ 0.43 $10^7$ km$^2$). The variability is characterized by increased spectral power on precession (~ 21 kyr), obliquity (~ 41 kyr), and $CO_2$/ice-sheet (~ 80-120 kyr) frequencies across the 3 Ma (Figs. 14b-c), consistent with the combined effect of GHG and ice-sheet forcing.

Sea ice extents both in the NH and SH exhibit a pronounced post-MPT climate shift. Considering the close relationship between sea ice and mid-latitude westerlies via the modulation of the meridional temperature gradient and atmosphere's baroclinicity (Timmermann et al., 2014a;Lamy et al., 2019;Menviel et al., 2010), we expect that the post-MPT climate shift would be more robust in the NH than the SH, due to the stronger obliquity-scale frequency of atmospheric circulation in the NH (see Figs. 12b and c). However, the obliquity signal of sea ice extent is more pronounced in the SH. The discrepancy between both hemispheres could be primarily explained by different geographical feature: in the NH sea ice is mostly limited to the Arctic Ocean, whereas in the SH it is unconstrained. This suggests an interesting feature of inter-hemispheric asymmetry for global climatic changes. For example, for high obliquity and high atmospheric $CO_2$ levels (positive PC1 values), the heat is transported equatorward primarily by means of the atmosphere in the NH or the ocean in the SH (see Fig. 11e). In the SH, the increased high-latitudes insolation can melt more sea ice and the Southern Ocean then transports more heat through the mean equatorward Ekman transport (Morrison et al., 2016). This would reflect an increasing role of sea ice on Southern Ocean heat transport after the MPT.

**7 Discussion, summary, and outlook**

We presented results from the first quasi-continuous Pleistocene simulation, conducted with a CGCM, and using time-evolving forcings of the orbital parameters, GHG concentrations, and NH ice sheets. Our simulation represents well the observed orbital-scale shifts not only in the regional mean temperature changes but also in hydro-climate and large-scale westerly variabilities controlled by Earth's





meridional temperature gradients. We identified a simple rectification mechanism by which eccentricity-modulated precessional cycles in precipitation can introduce mean state variability with periods of 100 and 405 kyrs in the tropical climate system. This finding may provide a simple framework for interpreting
405 kyr eccentricity variability in long-term paleoclimatic records (Kocken et al., 2019;Nie, 2018). Simulated ENSO variability is also driven by eccentricity-modulated precessional cycles which cause an anomalous seasonal cycle forcing that interacts with ENSO. Tropical mean state changes, which are in large parts influenced by $CO_2$ and ice-sheet forcing, show little impact on ENSO variability and flavors.

To provide a larger context for our simulation we focused on the temporal variability of global
heat transport. Two dominant modes govern the global energy transport across the 3 Ma Pleistocene: a "tropical convergence mode" related to equator-to-pole temperature gradient and an "inter-hemispheric shift mode", which is linked to the inter-hemispheric temperature difference. These processes are accompanied by characteristic changes in both atmosphere and ocean circulations driven by westerly wind changes, the AMOC, and sea ice albedos. The two MHT modes reveal a robust regime shift for the
pre- and post-MPT periods, which is most strongly pronounced for the second MHT mode. During the post-MPT glacial peaks, there is an increasing probability (from 15.3% to 47%) of anomalous poleward heat transport (i.e., negative PC1; colder poles than tropics) and southward shifted heat transport (i.e., negative PC2; colder SH than NH), which could contribute to cooling in the NH climate and potentially an intensification of glacial conditions. We emphasize that the regime shift in the major MHT modes
plays a pivotal role in reshaping the inter-hemispheric exchange of energy, thereby contributing to the glacial interhemispheric temperature heterogeneity and interglacial homeostasis during the late Pleistocene.

According to our analysis, both hemispheres (NH and SH) are important in controlling the Pleistocene global energy redistribution. Considering the role of AMOC changes in global energy
transport (Frierson et al., 2013;Yu and Pritchard, 2019) we find in our simulation a more pronounced role contribution of the atmospheric circulation changes to orbital-scale (> 20 kyr) variations in the MHT, as compared to the ocean. In the Southern Ocean, the increased variance in sea ice after the MPT may have further contributed to climate-carbon cycle feedbacks, which are not explicitly resolved in our simulation because we use prescribed $CO_2$ forcing. In a more realistic setting which involves an interactive carbon



cycle, the two MHT modes and shifts in sea-ice may have influenced the outgassing of $CO_2$ from the ocean to atmosphere on glacial/interglacial timescales (Stein et al., 2020). For example, negative signs of PC1 and PC2 (related to Southern Ocean cooling) would amplify the Southern Ocean $CO_2$ sequestration, because of the increased sea ice-carbon cycle feedback and increased sinking of carbon-rich deep waters (Stott et al., 2007;Stein et al., 2020) (see Fig. S6 in Supplement). Thus, we hypothesize that the increased

occurrence of same-sign PCs anomalies after the MPT may have helped re-enforce the glacial carbon cycle response, whereas prior to the MPT, the opposite signs of PCs may contribute partially to the compensation between $CO_2$ sequestration and $CO_2$ outgassing in the SH. Therefore, the global energy redistribution could have served as feedback in triggering glacial carbon cycle response during the late Pleistocene. Our simulation provides paleoclimate support for observational and modeling studies that

link changes in atmosphere and ocean circulations to the redistribution of the global energy transport. Together, our analysis suggests a coherent interpretation of 3 Ma Late Pliocene/Pleistocene climate changes in large-scale circulations and global energy transport.

In this last paragraph, we will put the simulated global mean surface temperature variability over the past 3 Ma into the context of recent and future anthropogenic climate change. To this end, we ran

additional simulations with the CESM1.2 model using the historical forcings (1850-2015) and Representative Concentration Pathway (RCP) 2.6, 4.5 and 8.5 GHG emission scenarios (2016-2100) (Fig. 15). We find that the typical late Pleistocene glacial/interglacial temperature range of 4-6 °C is comparable in amplitude to the RCP8.5 GHG warming projections over the next 7 decades (~ 5.0 °C) (Fig. 15). In terms of warming rates (°C 100 $yr^{-1}$), however, the anthropogenic projections exceed the natural

variability by almost 2 orders of magnitude. This is likely to push global ecosystems way outside the range of temperature stress that they may have experienced naturally, at least within the last 3 Myrs.

The data from our quasi-continuous CESM1.2 3Ma simulation are available on the OpenDAP and LAS server https://climatedata.ibs.re.kr and we hope that further analyses of these runs will help in elucidating the mechanisms of past climate shifts and in the interpretation of paleo-climate proxies.




## Author Contributions

AT and K-SY designed the study. K-SY conducted the CESM1.2 3Ma simulation and wrote the initial manuscript draft and produced all figures. All authors contributed to the interpretation of the results and to the improvement of the manuscript.

## Acknowledgements

This study was supported by the Institute for Basic Science (project code IBS-R028-D1). The CESM1.2 simulations were conducted on the IBS supercomputer "Aleph", 1.43 peta flops high-performance Cray XC50-LC Skylake computing system with 18,720 processor cores, 9.59 PB storage, and 43 PB tape archive space. We also acknowledge the support of KREONET.

## Data and Code availability

The transient Pleistocene 3 Ma model data will be made available on the openDAP ICCP climate data server https://climatedata.ibs.re.kr. Codes used in this study are available from the authors upon request. The code and resources for paleo-climate configuration are also available from the CESM1.2 series Paleo-climate toolkit website (https://www.cesm.ucar.edu/models/paleo). Interactive Data Language (IDL) version 8.8 was used to generate all figures.

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

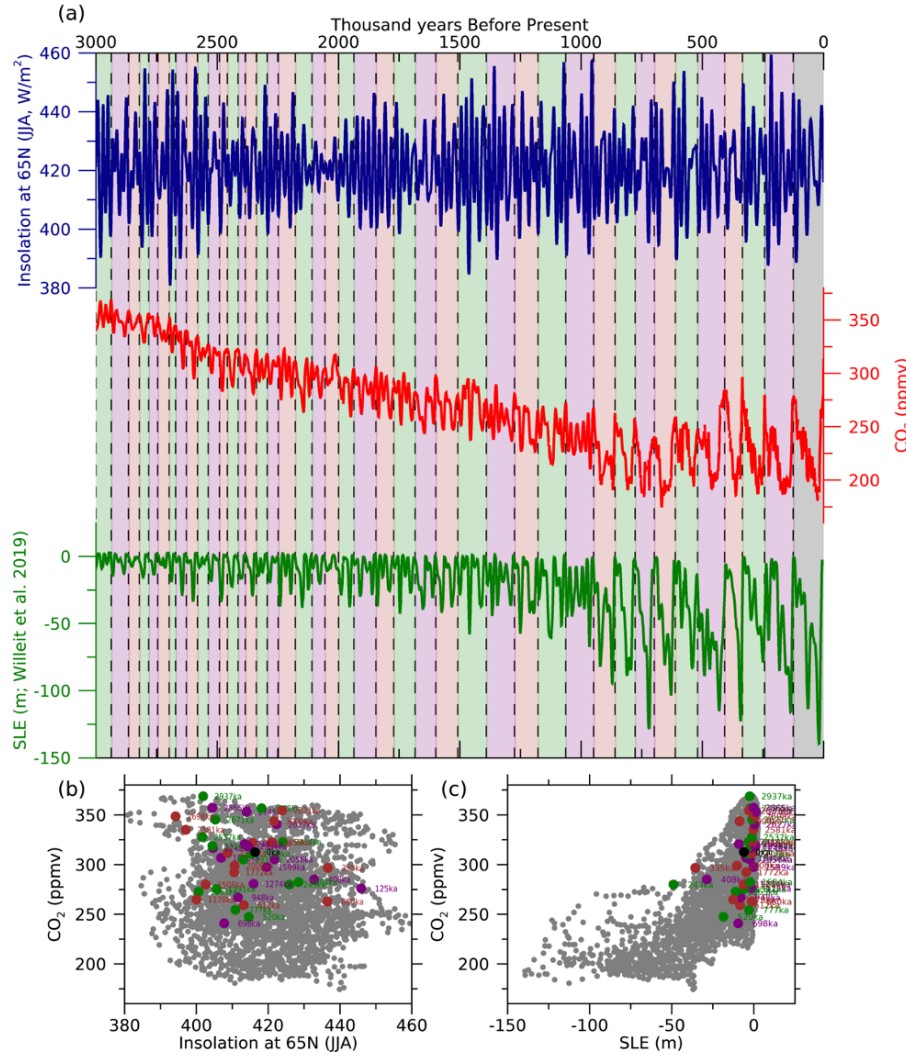


**Figure 1: "3 Ma" ensemble simulation strategy.** (a) Transient climate variability of summer insolation at 65°N, $CO_2$, and ice volume in sea level equivalent (SLE, m) from Willeit et al. (2019). Each colour shading indicates the respective period of 42 individual ensemble simulations. (b-c) Scatter plot of (b)



summer insolation versus $CO_2$, and (c) SLE versus $CO_2$. Each ensemble run was initiated at interglacial
peaks (e.g., 125ka and 243ka) of having higher $CO_2$ and lower ice volume, as displayed by coloured dots.

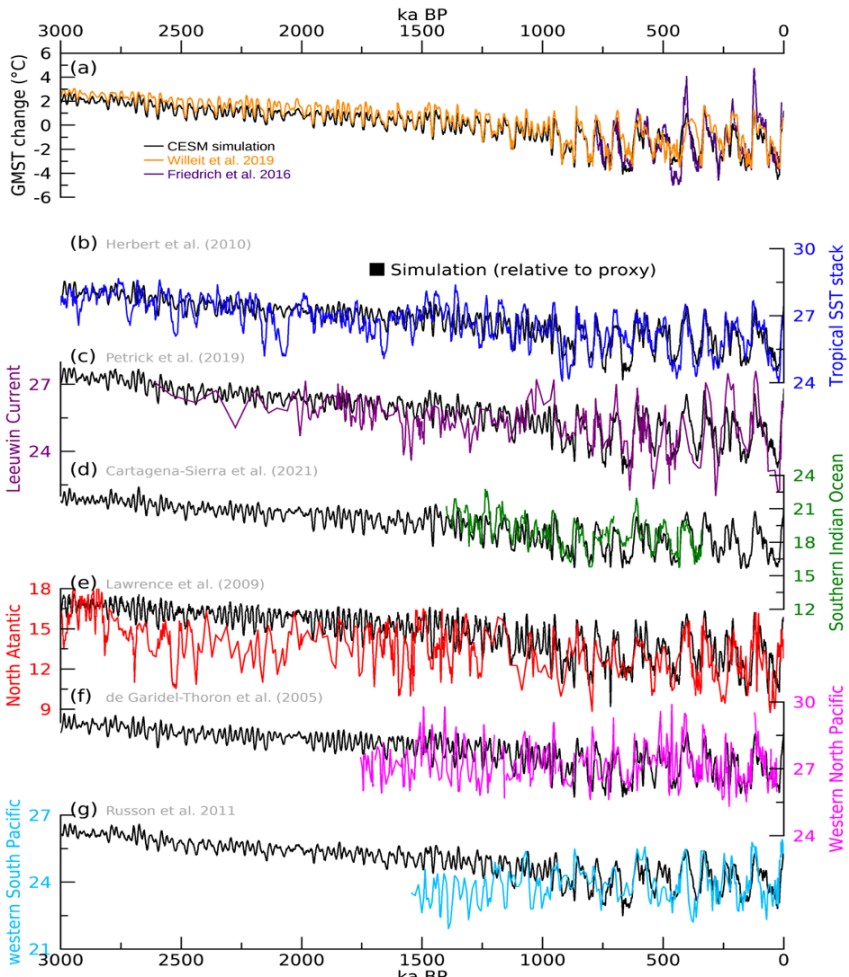

**Figure 2: CESM 3 Ma simulation and climate sensitivity.** (a) The simulated global mean surface air
temperature anomalies relative to the 3 Ma period (black). Previous global mean temperature estimates
are displayed in different colors: orange (Willeit et al., 2019) and violet (Friedrich et al., 2016). (b-c) SST
comparison between paleo-proxy data and CESM simulation (black): (b) tropical SST stack (Herbert et
al., 2010) and tropical SST simulation (averaged over 5°S-5°N, unit:°C), (c) Leeuwin Current from



Integrated Ocean Drilling Program (IODP) site U1460 (Petrick et al., 2019). (d) Southern Indian Ocean
from sediment core MD96-2048 (Cartagena-Sierra et al., 2021). (e) North Atlantic from Ocean Drilling
Program (ODP) 982 (Lawrence et al., 2009), (f) Western North Pacific from International Marine Global
Change Study (IMAGES) core MD97-2140 (de Garidel-Thoron et al., 2005), and (g) western South
Pacific from MD06-3018 (Russon et al., 2010). The proxy locations are also displayed in Fig. S2 in the
Supplement.


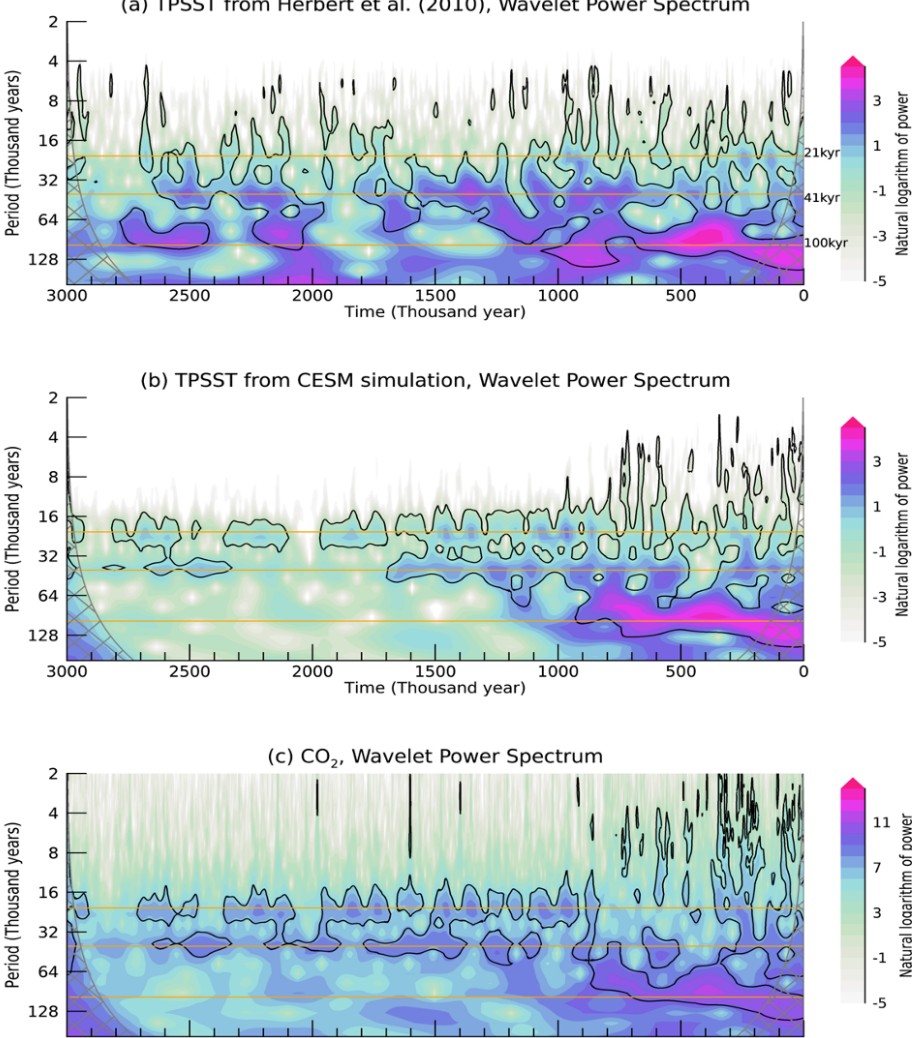



**Figure 3: Wavelet power spectrum of tropical SSTs and CO₂.** The wavelet power spectrum of (a-b) tropical SST obtained from (a) alkenone data and (b) CESM simulation and of (c) CO$_2$ concentration. The black contour indicates the value significant at the 95% confidence level. The horizonal orange lines show 21 kyr (precession), 41 kyr (obliquity), and 100 kyr (eccentricity) periods.

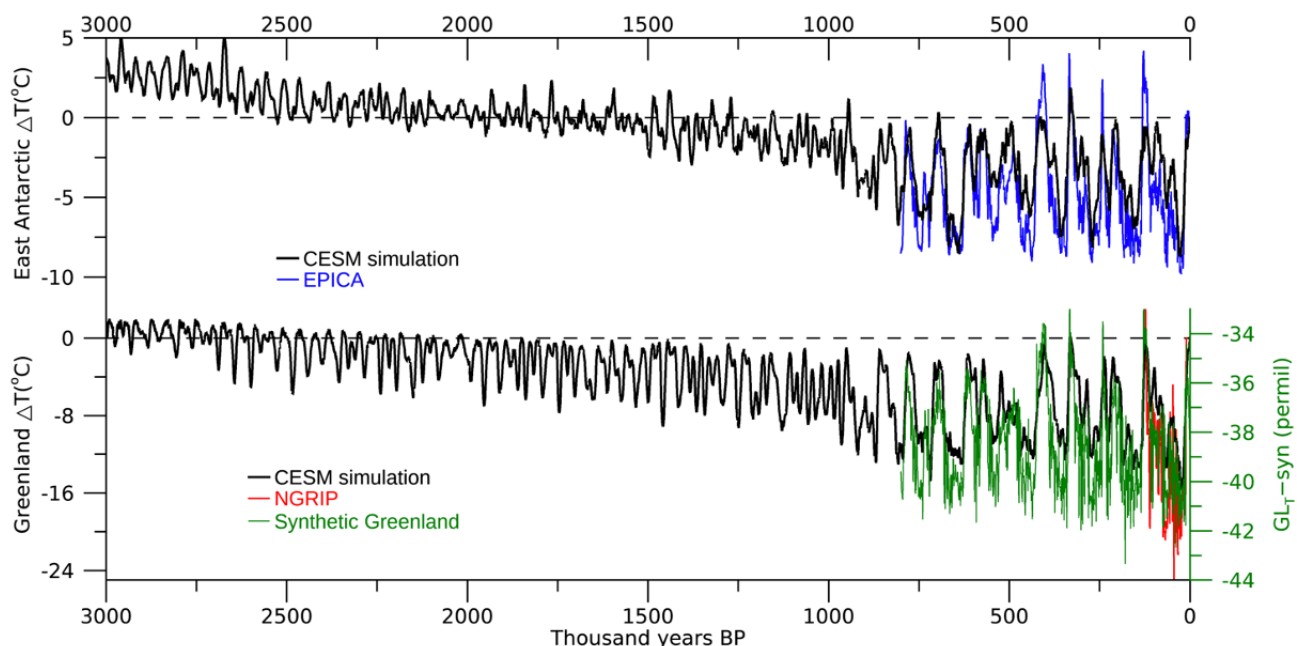

**Figure 4: Polar climate changes.** The simulated surface air temperature anomalies relative to 0 ka in east Antarctica (Upper; 75°S, 123°E) and in Greenland (bottom; 75°N, 42°W) (blue). Proxy-based estimations, i.e., Antarctic temperature from European Project for Ice Coring in Antarctica (EPICA) Dome C ice core (Jouzel et al., 2007; blue in upper), North Greenland Ice Core Project (NGRIP) Greenland temperature (Kindler et al., 2014; green in bottom), Synthetic Greenland temperature (Barker et al., 2011; red in bottom), are also displayed in different colors.

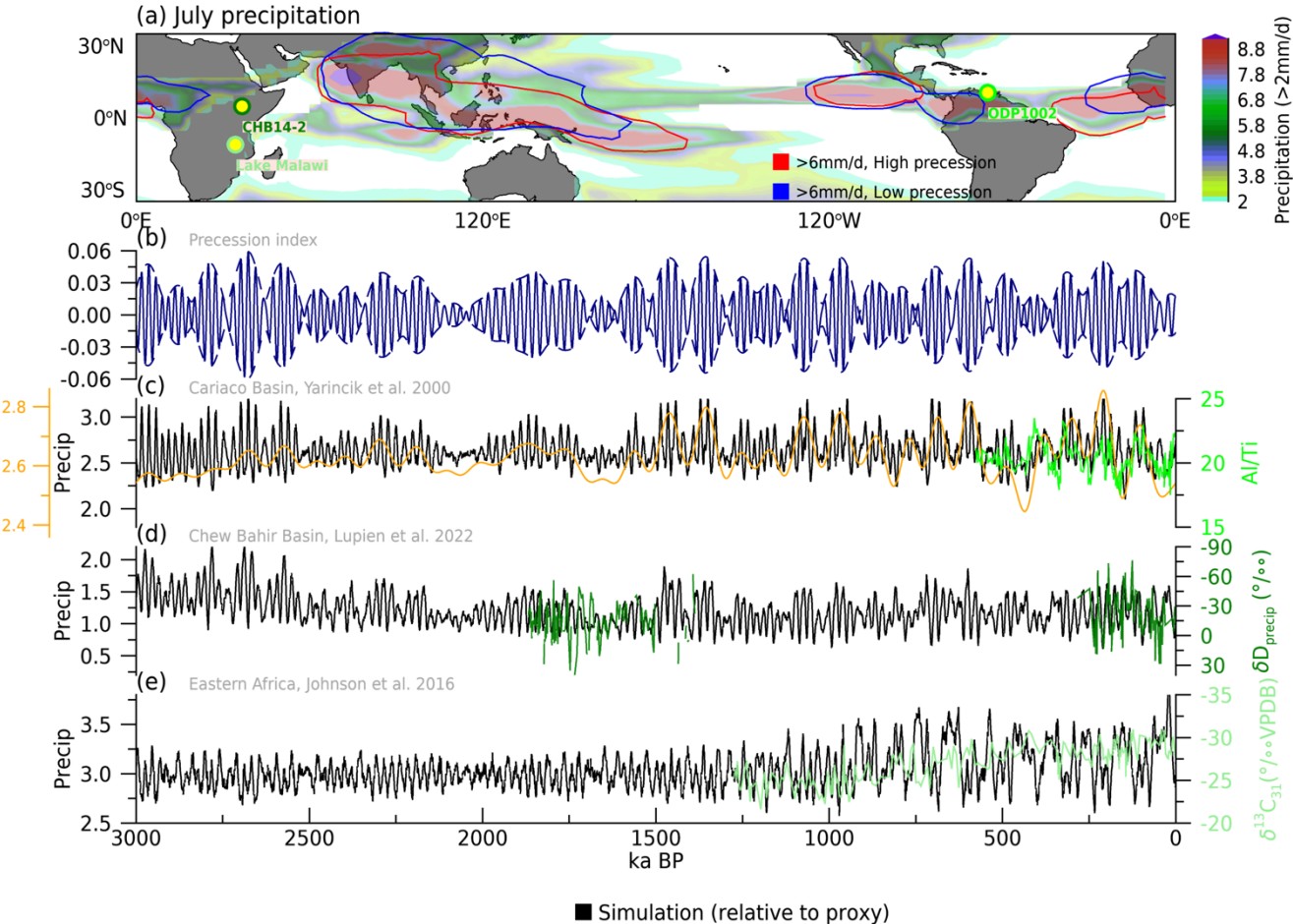

**Figure 5: ITCZ changes.** (a) Simulated July precipitation pattern (unit: mm day⁻¹) averaged for the 3 Myr period. Each proxy location is displayed in different colors. The tropical precipitation regions greater than 6 mm day⁻¹ are displayed for comparison between the phases with high precession (red) and low precession (blue). (b) Precession index ($e \cdot \sin\varpi$ where $e$ is the eccentricity and $\varpi$ is the moving longitude of the perihelion). (c-e) Precipitation comparison between the proxy and simulation (black): (c) Cariaco Basin from ODP 1002 Al/Ti ratio (Yarincik et al., 2000), (d) Chew Bahir Basin and Turkana basin from CHB14-2and WTKB $\delta D_{precip}$ (Lupien et al., 2022), and (e) Eastern Africa from Lake Malawi $\delta^{13}C_{31}$





(Johnson et al., 2016). The orange line in (c) indicates the 80 kyr low-pass filtered Cariaco Basin precipitation in simulation.

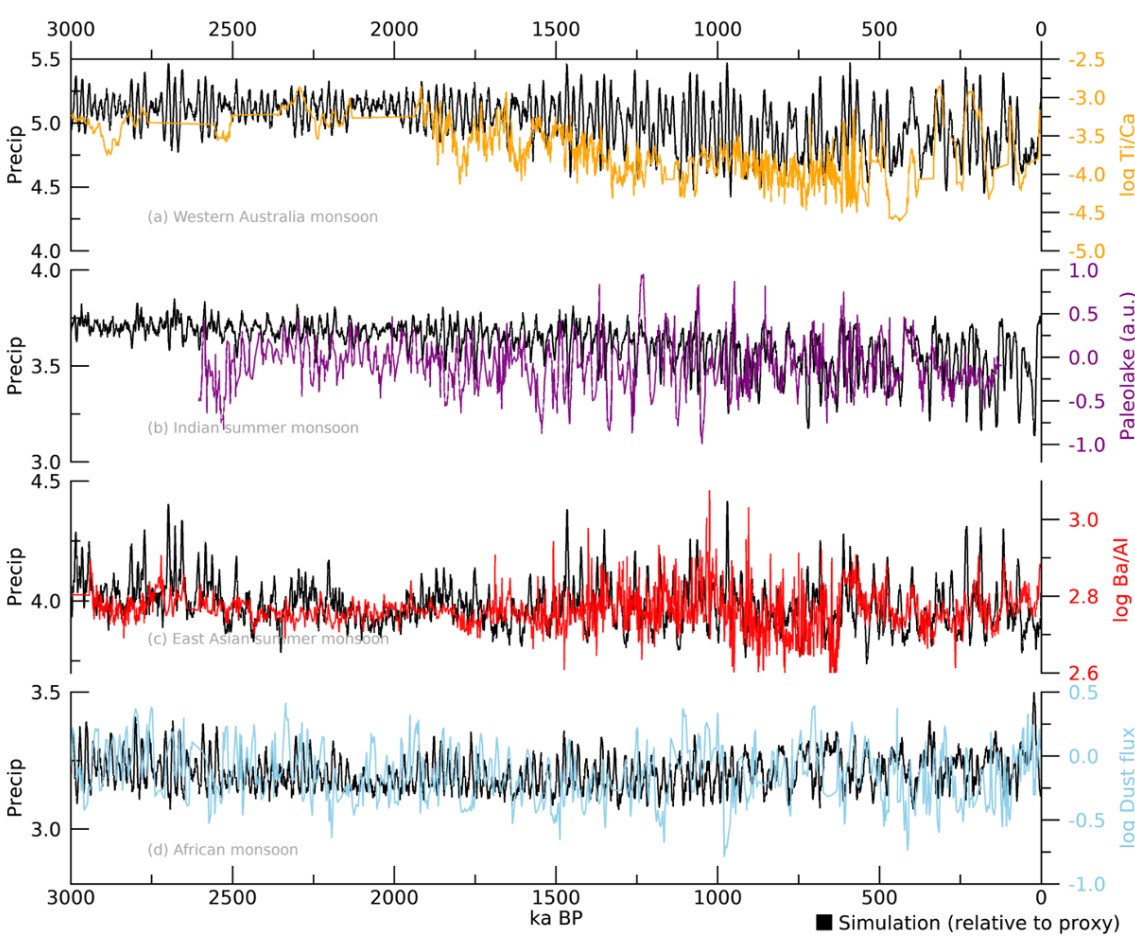

**Figure 6: Global hydroclimate changes.** Precipitation (unit: mm d⁻¹) comparison between the proxy and simulation: (a) Western Australia monsoon from simulation averaged over 20°S-5°S, 110°E-150°E (black) and from proxies of IODP site U1460 log Ti/Ca (Petrick et al., 2019; orange); (b) Indian summer monsoon from simulation averaged over 10°N-30°N, 70°E-105°E (black) and from proxies of Lake Heqing (An et al., 2011; purple); (c) East Asian summer monsoon from simulation averaged over 20°N-40°N, 110°E-130°E (black) and from proxies of ODP 1146 log Ba/Al (Clemens et al., 2008; red); (d) western African monsoon from simulation averaged over 25°S-5°S, 25°E-50°E (black) and from proxies of ODP 659 log



Dust flux (Tiedemann et al., 1994; sky blue). The proxy locations and respective monsoon domains are
presented in Fig. S2 in the Supplement.

**Figure 7: Global westerly wind changes.** (a) Simulated zonal wind pattern at 500hPa averaged for the
3 Myr period. Proxy locations and Pacific/Atlantic maximum wind regions at 40°N and 40°S are
displayed in different colors. (b-e) Westerly comparison between the paleo-proxy data and CESM
simulation (black): (b) North Atlantic wind from U1313 n-alkane flux (Naafs et al., 2012) and simulation
(averaged over 40°N, 60°W-33°W). (c) North Pacific wind from ODP882 (Martínez-Garcia et al., 2010)
and simulation (40°N, 140°E-170°E), (d) South Pacific wind from log Fe/Ca record in fluvial sediment



input (Lamy et al., 2019) and simulation (40°S, 120°W-90°W), and (e) South Atlantic wind from

805 ODP1090 dust flux (Martínez-Garcia et al., 2011) and simulation (40°S, 30°W-0°).

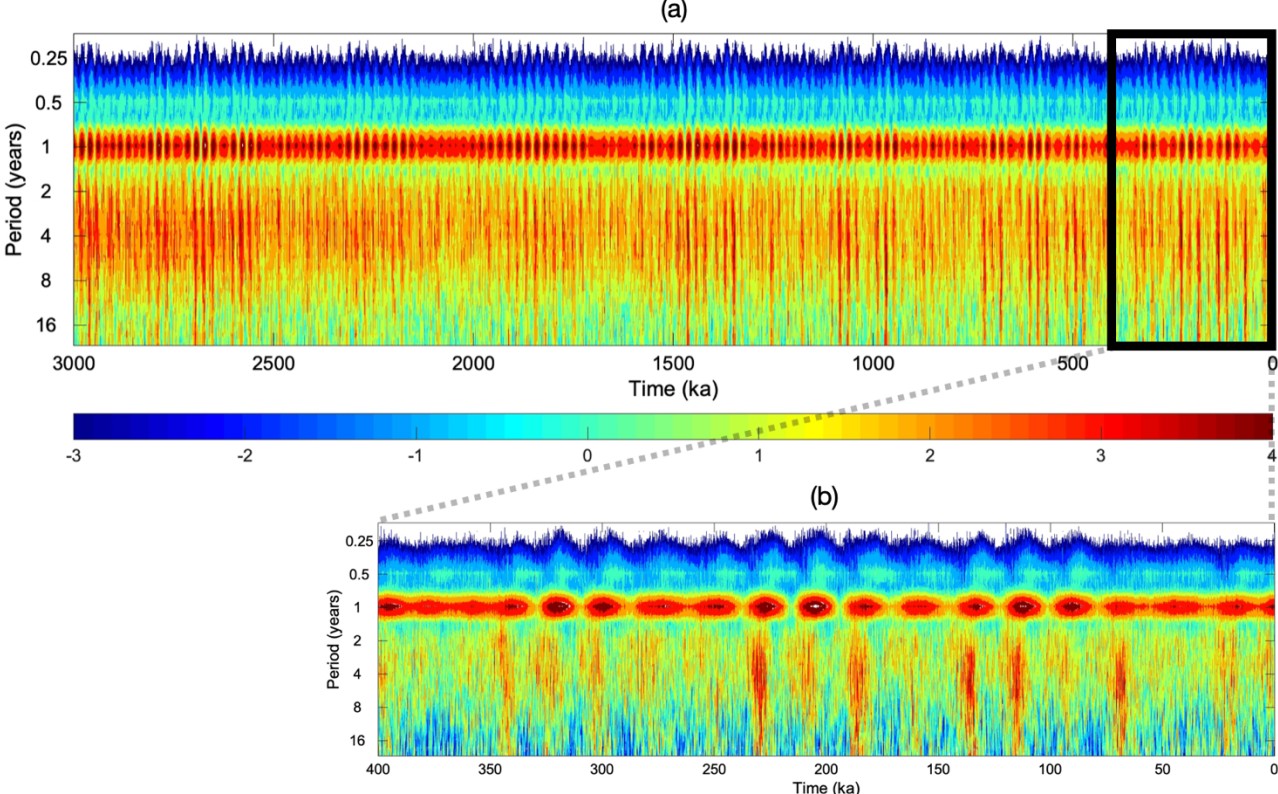

**Figure 8: Monthly ENSO spectrum.** Wavelet power spectrum (logarithmic variance) of Niño 3 SST (5°S-5°N, 150°W-90°W) during the period of (a) 3 Myr and (b) 400 kyr.





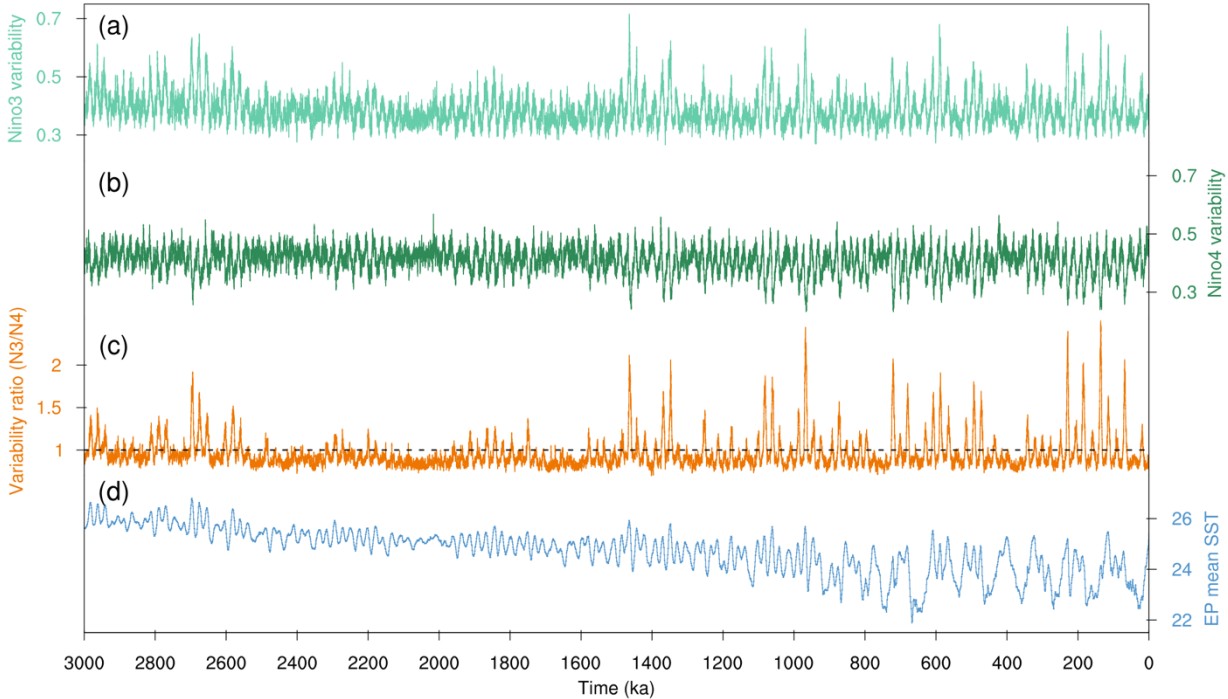

**Figure 9: Comparison of monthly Niño 3 and Niño 4 variabilities.** Timeseries of (a) Niño 3 SST index variability (°C), (b) Niño 4 SST index variability (°C), (c) ratio of Niño 3 variability to Niño 4 variability, and (d) 1000-year mean Niño 3 SST (°C).



**Figure 10: Global heat transport across the 3 Ma simulation.** (a) The meridional heat transport (MHT; unit: PW; black line), atmospheric MHT (AHT; red line) and oceanic MHT (OHT; blue line) averaged over the entire 3 Myr period (solid), 41 kyr period (i.e., before the Mid-Pleistocene Transition (MPT), about 1Ma; dot line), and 100 kyr glacial period (i.e., after the MPT; dashed line). The minimum and maximum range of MHT changes over the 3 Myr period is displayed by corresponding color shading. (b) Box-whisker plot (i.e., minimum, 25%, 50%, 75%, maximum range) of contribution of OHT to MHT (unit: %) at the latitude position of 60°S, 40°S, 15°S, 15°N, 40°N, and 60°N during the three different periods: the entire 3 Myr period (black), 41 kyr period (orange), and 100 kyr glacial period (brown). The contribution of AHT to MHT can be estimated by subtracting the contribution of OHT from 100%. (c) The time-latitude plot of MHT anomaly (unit: $10^{-1}$ PW) relative to the climatological mean. Positive MHT indicates the northward heat transport.



**Figure 11: Spatio-temporal modes of global heat transport variability.** (a-b) The principal component
(PC) timeseries associated with the first two leading modes of MHT anomalies. (c-f) The regressed
anomalies against the (left) PC1 and (right) PC2 variability: (c-d) zonally averaged top-of-atmosphere
net radiative flux (downward in positive; unit: W m$^{-2}$) and surface net heat flux (toward ocean in positive;
unit: W m$^{-2}$), (e-f) the total MHT (black), Oceanic MHT (OHT; blue), and Atmospheric MHT (AHT;
red) (unit: PW). Sky-blue and pink shading in (e-f) indicates the weakening and strengthening of
poleward total MHT, respectively.

835

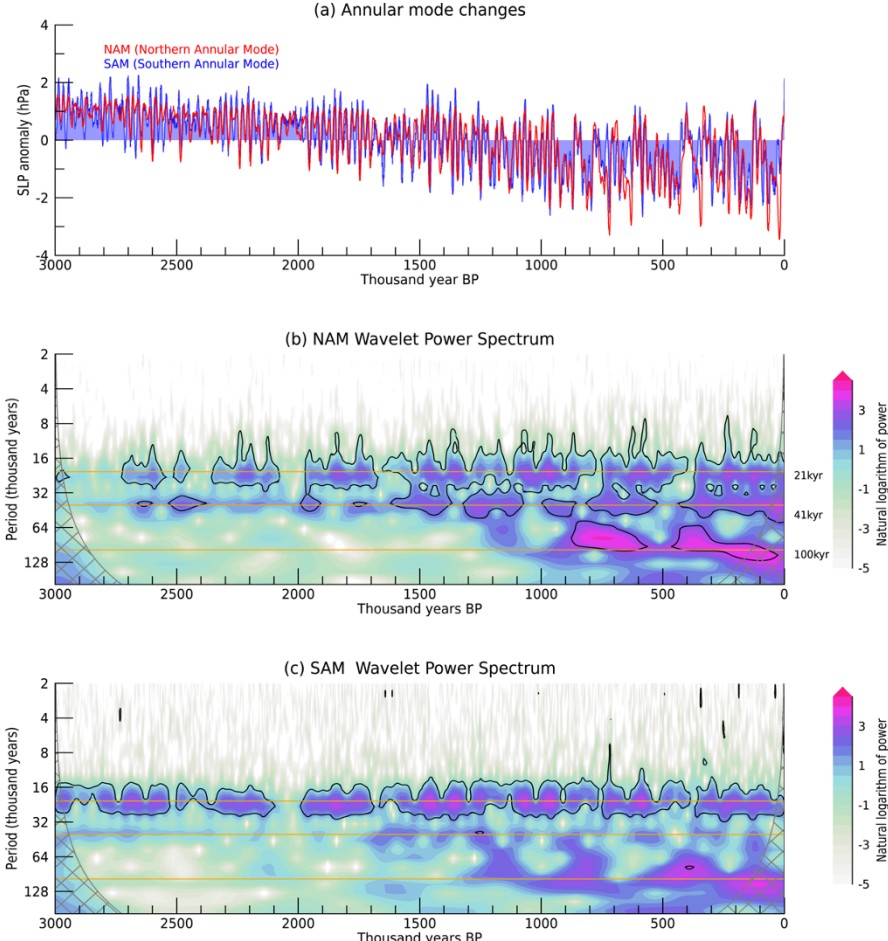

**Figure 12: Atmosphere circulation changes.** (a) The timeseries of Northern Annular Mode (NAM) and
Southern Annular Mode (SAM). Here, the NAM is defined as the difference in zonal mean SLP at 35°N
and 65°N (Li and Wang, 2003) and the SAM as the difference in zonal mean SLP at 40°S and 65°S
(Gong and Wang, 1999). (b-c) The wavelet power spectrum of (b) NAM and (c) SAM indices. The black
contour indicates the value significant at the 95% confidence level. The horizontal orange lines show 21
kyr (precession), 41 kyr (obliquity), and 100 kyr (eccentricity) periods.



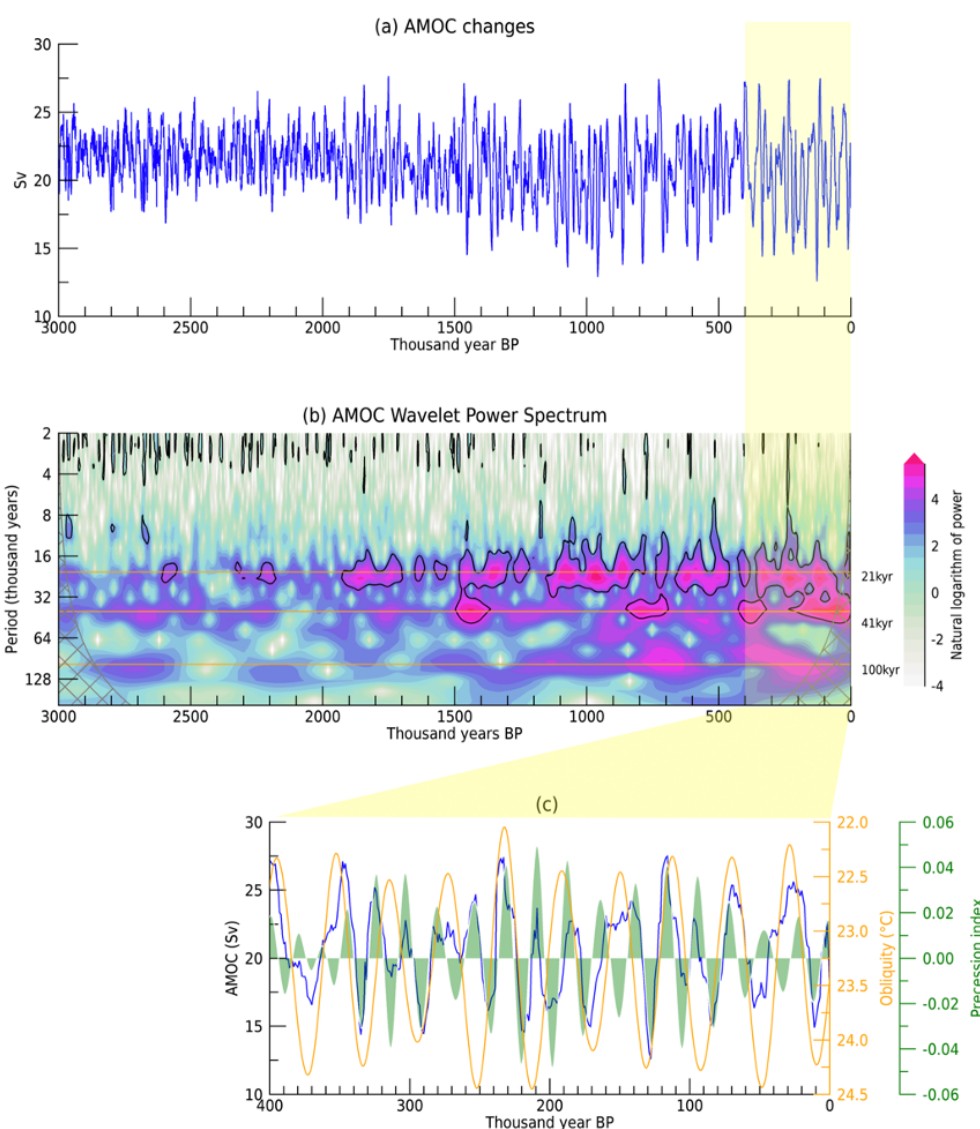

845

**Figure 13: Ocean circulation changes.** (a) The timeseries of Atlantic Meridional Overturning circulation (AMOC). Here, the AMOC amplitude is defined as the maximum meridional stream function below 500m and north of 28°N. (b) The wavelet power spectrum of AMOC amplitude. The black contour indicates the value significant at the 95% confidence level. (c) 400ka evolution of AMOC, precession and obliquity variability. The horizonal orange lines show 21 kyr (precession), 41 kyr (obliquity), and 100 kyr (eccentricity) periods.





**Figure 14: Sea ice changes.** (a) The timeseries of sea ice extent over the Northern Hemisphere (NH, red) and Southern Hemisphere (SH, blue). (b-c) The wavelet power spectrum of (b) NH sea ice and (c) SH sea ice. The black contour indicates the value significant at the 95% confidence level. The horizonal orange lines show 21 kyr (precession), 41 kyr (obliquity), and 100 kyr (eccentricity) periods.

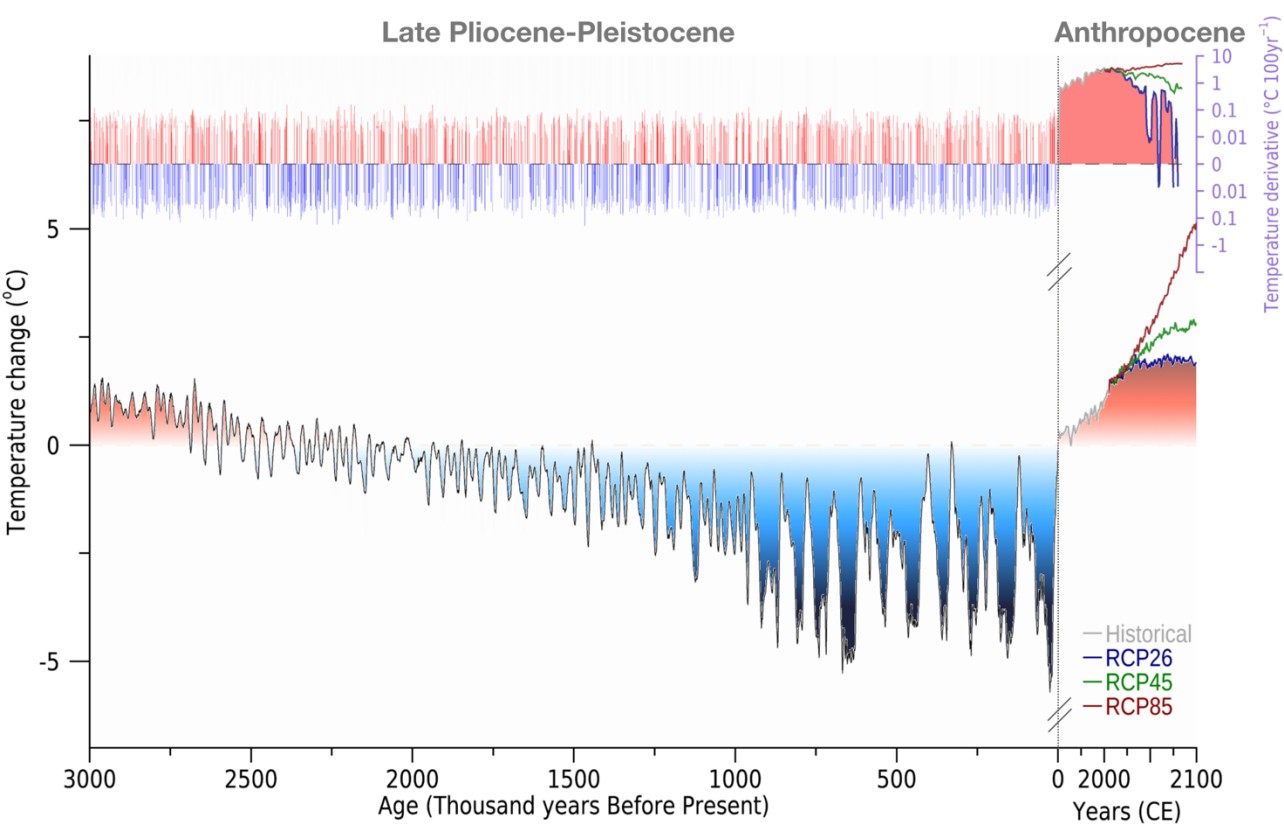

**Figure 15: Past-to-future warming context.** The simulated global mean surface temperature anomalies relative to 0 ka for 3 Ma and CESM1.2 simulations using historical forcings (1850-2005) and greenhouse warming scenarios following the Representative Concentration Pathway (RCP) 2.6, 4.5, and 8.5 (2016-2100). The upper panel shows the corresponding rate of global mean temperature change (°C 100 yr$^{-1}$; right axis).

