# Peer review of "A transient CGCM simulation of the past 3 million years"

_Climate of the Past, 2023_

## Author Comment (AC1)

[Research Article # cp-2023-34]: "A transient CGCM simulation of the past 3 million years" by Kyung-Sook Yun, Axel Timmermann, Sun-Seon Lee, Matteo Willeit, Andrey Ganopolski, and Jyoti Jadhav

We thank the reviewers for their constructive and helpful comments. We carefully revised the manuscript "A transient CGCM simulation of the past 3 million years" and provide a point by point reply to the individual comments below.

**Reply to the comments of Reviewer #1**

**General Remarks:**
*The manuscript presents an unprecedented effort to simulate the Pleistocene until present and beyond. It is the first quasi-continuous simulation using a comprehensive coupled general circulation model over a time span as long as three million years. This approach allows the authors to fill a gap between ultra-long simulations using models of intermediate complexity (EMIC) and time slices or short transient runs with fully coupled atmosphere -ocean or Earth system models. The CGCM CESM1.2 used in this study is well suited. Although of coarse resolution, it has been used since many years and has been shown to give satisfying representation of past and present climate states and variability (e.g. ENSO).*
*The technical design of the experiment is impressive and well thought-through. The authors make use of a well-established acceleration technique I combination with running chunks of the simulation in parallel.*
*The authors show that the model reproduces the climate evolution in accordance with reconstructions and earlier EMIC simulations (from which, however, the forcing driving the CGCM was derived). More importantly, the authors do a very good job highlighting the added value of the CGCM simulation: the ability to look how atmospheric dynamics and climate variability change under orbital and greenhouse gas forcing. In particular, I appreciate the investigation on the meridional heat transports in ocean and atmosphere. Here the authors come up with very interesting new results.*
*On a more critical note, the authors could discuss in more detail shortcomings and limits of their experiment. For example, the static LGM topography and land-sea mask. Another important limitation is that there is no coupling of ice-sheet changes and fresh-water forcing of the ocean. This is mentioned in the AMOC section, but it would be more important to discuss the constraints on the meridional heat transports in the ocean under AMOC breakdown and recovery.*
*Overall, the manuscript is mature and written in a concise and clear manner. The figures are of good quality and convey the message well. I have only identified a very few minor issues (see below).*

*I therefore recommend accepting the paper with (very) minor revision.*

[Ans] We are grateful to the reviewer for very constructive comments which were helpful to improve our revised manuscript. It is correct, that our simulation does not include changing land-sea configurations, interactive ice-sheets and associated freshwater changes. Moreover, the model does not represent dynamic vegetation changes, dust and aerosol effects, and climate-carbon cycle feedbacks, which are likely to play a role in shaping the Pleistocene climate trajectory. For example, we used the constant LGM land-sea mask throughout the entire

simulation, probably leading to biases in regional temperature and hydroclimate (e.g., Cao et al., 2019). In addition, in the absence of ice-sheet fresh-water forcing, the amplitude of AMOC variability will be suppressed on millennial, but probably also on orbital timescales and this could alter the contribution of OHT to the MHT. In our revised manuscript we provide a more extensive discussion of the limitations of our experimental set-up.

We also addressed the reviewer's minor comments, as described below.

**Minor comments:**
***Q. 1.** Line 175: could you comment on the discrepancy between "synthetic Greenland" and the simulation, i.e. the much larger amplitude of the reconstruction?*

[Ans] The "synthetic Greenland" temperature is based on a statistical reconstruction which uses $\delta^{18}O$ of ice in Greenland and water isotopes from Antarctica (to extend the record beyond MIS5a). To compare this reconstruction quantitatively with our simulation, we use the scaling between NGRIP temperatures over the last 120 kyr (Kindler, 2014) and NGRIP $\delta^{18}O$ of ice (NGRIP members, 2014) ranging between -32 and -46 permille and the synthetic Greenland $\delta^{18}O$ record. We re-scaled the figure accordingly and modified the description in the revised text [line 841-843]. Using this scaling we can see that the simulated temperatures (see Ref_fig1) show a reasonable agreement with the corresponding temperature reconstructions from Greenland, except for the millennial-scale variability, which is not captured by the 3 Ma simulation.

[Figure]

**Ref_fig.1 Comparison between Greenland temperature and $\delta^{18}O$ of ice over the 125ka.** The simulated annual mean air temperature anomalies relative to 0 ka in Greenland (75°N, 42°W) (black; left axis). Proxy-based estimations, i.e., NGRIP Greenland temperature (Kindler et al., 2014; red color; left axis), NGRIP $\delta^{18}O$ of ice (NGRIP members, 2004; blue color; right

axis), and Synthetic reconstruction of Greenland temperature (Barker et al. 2011; green color, right axis), are also displayed in different colors.

NGRIP members: High-resolution record of Northern Hemisphere climate extending into the last interglacial period, Nature, 431, 147–151, doi:10.1038/nature02805, 2004.

*Q. 2. Line 214: the "intensification in amplitude after the MPT" in the East Asian monsoon: this looks more like a gradual increase from 1500 kaBP. Could you be a bit more qauntiative here, e.g. by showing running standard devitations?*

**[Ans]** The point is well taken. We calculated the 31-kyr window running standard deviation of simulated precipitation and hydroclimate proxies (Ref. fig2). In the East Asian monsoon region, there is indeed a gradual increase in variability in some monsoon systems from 1.5 Ma. We added this figure as Fig. S3 in the Supplement and modified the revised text accordingly [line 236-244].

[Figure]

**Ref_fig.2 Changes in monsoon variability amplitude.** Sliding standard deviation of precipitation (unit: mm d$^{-1}$) and proxies over 31-kyr time window: (a) Western Australia monsoon from simulation averaged over 20°S-5°S, 110°E-150°E (black) and from proxies of IODP site U1460 log Ti/Ca (Petrick et al., 2019; orange); (b) Indian summer monsoon from simulation averaged over 10°N-30°N, 70°E-105°E (black) and from proxies of Lake Heqing (An et al., 2011; purple); (c) East Asian summer monsoon from simulation averaged over 20°N-40°N, 110°E-130°E (black) and from proxies of ODP 1146 log Ba/Al (Clemens et al., 2008;

red); (d) western African monsoon from simulation averaged over 25°S-5°S, 25°E-50°E (black) and from proxies of ODP 659 log Dust flux (Tiedemann et al., 1994; sky blue).

*Q. 3. Line 294: AMOC and ocean heat transports (see above)*

**[Ans]** We agree with the reviewer that by not including ice-sheet fresh-water forcing in our model set-up, the amplitude of AMOC variability will be underestimated which will also impact the variations in OHT and their contribution to the MHT. A more detailed discussion of the limitations of our modeling approach (AMOC variability being one of them) is now included in our revised manuscript [line 473-483].

*Q. 4. Figure 2: caption more precise: relative to the mean over the entire 3 Ma.*

**[Ans]** We modified the figure caption accordingly [line 818].

*Q. 5. Figure 9: caption more precise: standard deviation over sliding time window*

**[Ans]** We modified the figure caption accordingly [line 898].

*Q. 6. Figure 10: include information how the characteristic modulation (labels in fig. 10 c) have been derived*

**[Ans]** For example, before the MPT, there are strong signals of obliquity for most latitudes and precession in the tropics and NH mid-latitudes, whereas after the MPT, we can see an increasing role of $CO_2$ on the MHTs. We provide an updated description in the revised text [line 325-327].

---

## Author Comment (AC2)

[Research Article # cp-2023-34]: "A transient CGCM simulation of the past 3 million years" by Kyung-Sook Yun, Axel Timmermann, Sun-Seon Lee, Matteo Willeit, Andrey Ganopolski, and Jyoti Jadhav

We thank the reviewers for their constructive and helpful comments. We carefully revised the manuscript "A transient CGCM simulation of the past 3 million years" and provide a point by point reply to the individual comments below.

**Reply to the comments of Reviewer #2**

**General Remarks:**
*This paper describes the first transient CGCM simulation of the last 3 Myrs, with a forcing acceleration factor of 5. The simulation is compared with proxy observations in various climate regimes and in various climate variables. This is a tremendous effort and believe the simulation will be a great resource for the community in future studies. The paper is well written and should be published.*

**[Ans]** We appreciate the reviewer's comments and suggestions. The manuscript has been revised according to the reviewer's comments, as listed below.

**Minor comments:**
**Q. 1.** *To better quantify the bias caused by the 5x acceleration, it will be good to have a comparison with a transient simulation without acceleration for one time period (say, about a precessional cycle length of about 20kyr).*

**[Ans]** We agree with the reviewer that potential biases may occur due to the use of the acceleration technique, in particular in the deep ocean and in high latitudes. The detailed comparison with a range of proxies, however, suggests that at least for near-surface variables the resulting biases or delays are likely very small. Moreover, according to Lunt et al. 2006, who ran an EMIC for the last 30,000 years with different acceleration factors (1, 2, 5, 10), and Timm and Timmermann (2007) and Varma et al. (2016) who ran LOVECLIM and CCSM3 with acceleration of 1 and 10 for different orbital-scale forcings, respectively, significant biases are to be expected for acceleration 10 for deep ocean temperatures and surface conditions in high-latitude regions where the climate is closely connected to the deep Ocean. Our factor 5 acceleration is a good compromise between minimizing distortions, delays and biases due to acceleration (see Figure 8 in Lunt et al. 2006), while at the same time maximizing computational performance. Unfortunately, running another 20,000 year unaccelerated run (300 years per day) with CESM1.2 would take another 2.5 months of computing time on our supercomputer, which is currently not available. Given, the large computational effort and the fact that such runs have already been conducted previously with EMICs (and we don't expect fundamentally different results with CESM1.2), we have refrained from launching a new unaccelerated run, but instead provide a more detailed discussion of the benefits and disadvantages of the method, as described in recent studies (Lunt et al. 2006, in particular for acceleration 5) [line 116-129].

Lunt, D. J., Williamson, M. S., Valdes, P. J., Lenton, T. M., and Marsh, R.: Comparing transient, accelerated, and equilibrium simulations of the last 30 000 years with the GENIE-1 model, Clim. Past, 2, 221–235, doi:10.5194/cp-2-221-2006, 2006.

*Q. 2. Clarify if the temperature discussed in the text and shown in the figures are all annual mean.*

**[Ans]** Yes. The temperatures discussed in the text and figures are based on all annual means. We revised the text accordingly [e.g., line 817].

*Q. 3. For the model data comparison of long time series, such as those in Figs.4-7, it will be good to have a parallel figure (say, to the right of the time series panel),which show two spectra, one before and one after the MPT. As it stands, it is becoming very hard to judge.*

**[Ans]** According to the reviewer's suggestion, we added a parallel figure showing two spectra of pre- and post-MPT in Figs. 4-7. Relevant descriptions are added in the revised text [e.g., line 196-198; line 242-244; line 256-257].

---

## Author Response (AR2)

[Research Article # cp-2023-34]: "A transient CGCM simulation of the past 3 million years" by Kyung-Sook Yun, Axel Timmermann, Sun-Seon Lee, Matteo Willeit, Andrey Ganopolski, and Jyoti Jadhav

We thank the editor for the comments. We revised the manuscript "A transient CGCM simulation of the past 3 million years" and provide a point by point reply to the individual comments below.

**Reply to the comments of Editor**

**Comments:**
*Q. 1.* *line 112: AMOC inserted without explaining acronym*

    **[Ans]** Revised [line 111].

*Q. 2.* *line 123: suppress comma*

    **[Ans]** Revised [line 123].

*Q. 3.* *line 138: typo (exchange parenthesis and comma)*

    **[Ans]** Revised [line 137].

*Q. 4.* *line 158: full stop missing*

    **[Ans]** Revised [line 152].

*Q. 5.* *line 241: rephrase "relatively produced well"*

    **[Ans]** Revised [line 240].

*Q. 6.* *line 389: please rephrase beginning of sentence*

    **[Ans]** Revised [line 388].

*Q. 7.* *line 477: suppress comma*

    **[Ans]** Revised [line 475].

*Q. 8.* *Finally, I generally fear font sizes in figures are very small. On the least numbers of dates in Figure 1b and 1c and references (e.g. Herbert et al (2010), etc) should be increased.*

    **[Ans]** Font sizes in figures were revised (Figs. 1, 2, 4,5,6, 7, and 9).